# Study of Alerting, Orienting, and Executive Control Attentional Networks in Bilingual and Monolingual Primary School Children: The Role of Socioeconomic Status

**DOI:** 10.3390/brainsci13060948

**Published:** 2023-06-14

**Authors:** Francesca Federico, Michela Mellone, Ferida Volpi, Margherita Orsolini

**Affiliations:** Department of Developmental and Social Psychology, Sapienza University of Rome, 78, Via Dei Marsi, 00185 Rome, Italy

**Keywords:** attention development, socioeconomic status, bilingualism, cognitive development

## Abstract

For decades, researchers have suggested the existence of a bilingual cognitive advantage, especially in tasks involving executive functions such as inhibition, shifting, and updating. Recently, an increasing number of studies have questioned whether bilingualism results in a change in executive functions, highlighting conflicting data published in the literature. The present study compared the performance of third-, fourth-, and fifth-grade bilingual and monolingual children on attentional and cognitive tasks. The participants were 61 monolingual and 74 bilingual children (*M* = 114.6 months; *SD* = 8.48 months) who were tested on two versions of the attention network task (ANT), with and without social stimuli, as well as tests investigating working memory, short-term memory, narrative memory, and receptive vocabulary. Data on families’ socioeconomic status and children’s reasoning abilities were also collected. The results showed that bilingualism and socioeconomic status affected attentional networks in tasks involving social stimuli. In tasks involving non-social stimuli, socioeconomic status only affected the alerting and executive conflict networks. Consistent with the literature, a positive relationship emerged between socioeconomic status and executive control in the context of social stimuli, and a negative relationship emerged between socioeconomic status and the alerting network in the context of non-social stimuli. Interestingly, neither socioeconomic status nor social attentional networks correlated with working memory. Therefore, although more investigations are required, the results suggest that differences in social contexts mainly affect attentional functions.

## 1. Introduction

Recently, interest in executive functions (EFs) has grown, with research showing that these abilities predict success in school and, more generally, life [1,2,3]. Attention control is an EF skill that is critical for operating efficiently in daily life, and gaps in this and other EFs have been shown to have severe consequences over the life course [4]. Additionally, longitudinal studies have found that attention deficit hyperactivity disorder (ADHD) is related to long-term impairment in EF, from childhood into adulthood [5,6,7]. Since some evidence suggests that bilingual individuals outperform monolingual individuals in response time (RT) (reflecting attention control) [8] and bilingualism is generally associated with a cognitive advantage, it seems relevant to investigate the cultural spaces that might provide the best learning experiences for children to inform the development of cognitive enhancement programs.

Recent research on bilingualism has produced mixed findings on the bilingual advantage. In more detail, a growing body of research is providing evidence that different bilingual profiles impact attentional skills differently. Since bilingual individuals must constantly control for the interference of another language, they may be disadvantaged in verbal tasks. On the other hand, bilingual individuals (particularly those living in a mixed-language environment) may benefit from the constant switching between languages, which offers them daily training to enhance their switching ability. Additionally, performance differences may arise from differences in ethnicity and socioeconomic status (SES). To address this, Morton and Harper [9] compared bilingual and monolingual children of identical ethnic backgrounds and SES. While the bilingual and monolingual children performed identically, children from higher SES families were advantaged compared to children from lower SES families. When the authors controlled for disparities in SES and ethnicity, the bilingual advantage in cognitive control was attenuated.

Further research is needed to investigate the factors (e.g., SES) that may distinguish different types of bilingualism to determine which provide an advantage and which do not.

### 1.1. Bilingualism and the Attentional Network

Studies demonstrating the benefits of bilingualism in the growth of cognitive processes have highlighted that the ability to speak two languages implies a suppression of the language that is not being spoken at any given time. The continuous activity of selecting and inhibiting language, according to the context, should imply better training of attentional networks, representing an advantage of bilingualism [10]. According to Bialystok [11], bilingual individuals also have an EF advantage because they are constantly training to conduct a conversation in context and require continuous access to working memory. Laurent and Martinot [12] found that school-aged children learning in a bilingual environment began to show the advantage of phonological awareness (one of the components of working memory) at the age of 9 years. Indeed, working memory is very important for conversation—particularly the ability of conversation partners to establish a joint perspective. Furthermore, in conversation, it is necessary to choose the appropriate language (and inhibit the other language) and observe what happens during the interaction (demonstrating cognitive flexibility).

Research has demonstrated that there is a close relationship between language skills and EFs [13], with Moreno et al. showing that music therapy training can improve both [14]. The language-based cognitive advantage is mainly reflected in tasks involving attentional networks [15,16]. According to Green’s joint activation model (1998) [17], a bilingual individual’s brain is always engaged in both languages, regardless of the language being spoken at any given time. Therefore, it is necessary for a general suppression mechanism to inhibit the activation of the non-target language. Monolingual individuals, in contrast, do not seem to require this suppression strategy. However, recent evidence shows that the bilingual advantage is more evident in executive attention than in inhibition.

Multilingualism tends to promote attentional disengagement, rather than overall inhibition, which is why it is visible in the sequential effect but does not demonstrate a traditional congruency effect [18]. Attentional disengagement is an aspect of executive attention that terminates attention to previous stimuli and reduces their influence on current processing. Some studies have found that differences between monolingual and bilingual children reduce when the groups are matched for general ability [19] and SES [9]. Other studies have suggested that monolingual–bilingual differences in cognitive abilities [19] are dependent on SES [9] and relate to processing efficiency. More recently, Naeem et al. [20] found that bilingualism had little effect on individuals with high SES but was essential for promoting processing speed among low-SES adults aged 18–30 years. Therefore, conflict adaptation paradigms may facilitate easier detection of the bilingual advantage than simple interference paradigms [18].

Bilingual individuals have significant experience disengaging attention from prior stimuli, due to the need to manage two languages. As a result, they are typically more immune to the impact of previous stimuli on current processing than monolingual individuals. In line with this, previous research has connected differences in conflict adaptation to differences in executive attention processes [21]. However, as Paap et al. described [22], “Bilingual advantages in executive functioning either do not exist or are restricted to particular and undetermined circumstances.” The authors noted that 80% of tests after 2011 failed to obtain results supporting the bilingual effect. They theorized that previous research on this topic might have been limited by an inability to control for various external factors (e.g., the experimental task and the participants’ SES and cultural and linguistic backgrounds) and small sample sizes.

Of note, low SES has been associated with lower cognitive performance [23]. Given the prevalence of the association between low SES and reduced vocabulary in bilingual individuals, several authors have emphasized the importance of analyzing this factor and monitoring its effects when a statistically significant difference is found between groups. Although many authors, e.g., [24,25], have expressed that statistically controlling for this variable is the most appropriate approach, others have claimed that SES is a specific feature of the population of interest [26] and that discrepancies in SES between monolingual and bilingual populations reflect variations in attentional disengagement [19,27,28]. A recent study by Orsolini et al. [29] investigated the role of working memory in reading comprehension between monolingual and bilingual Italian children. The results showed that working memory supported reading comprehension only indirectly. Indeed, written text comprehension is often described as a complex dual process involving working memory: multiple sources of information must be coordinated to constrain the development of a text representation, while relevant semantic content is kept alive in short-term memory. This dual process requires the direct involvement of the central executive. However, it does not appear to apply to elementary school students. A study found that monolingual children’s comprehension of an oral text was strongly related to their working memory skills; however, bilingual children showed no indirect effect of working memory on their reading comprehension [29].

Bilingual individuals are required to continuously handle two languages while speaking. This may have a general impact on their attentional networks. Some studies have reported that bilingualism is associated with better executive control. The most suitable model for analyzing the executive component of attention separately from other components (i.e., alerting and orienting) seems to be that of attentional networks. Costa et al. [14], testing young monolingual and bilingual adults with the attentional network task (ANT), showed that bilingual individuals were quicker at completing the task and more effective in their executive control and alerting networks. In particular, they benefited more from the presentation of alerting cues and were more efficient at resolving conflicting information. Additionally, compared to monolingual individuals, bilingual individuals had lower switching costs between trials. These findings suggest that bilingualism may impact young adults’ ability to develop effective attentional processes, precisely when that ability is expected to be at its peak.

Recent research by Park et al. [30] demonstrated that executive control was not affected by bilingual experience and was worse in children with developmental language disorders than in children with typical development. At least for children in this age range, dual language experience has been shown to have little impact on these skills. Tran, Arreondo, and Yoshida [31] observed that culture significantly impacted the development of alerting and executive control attentional networks in a sample of 3-year-old monolingual and bilingual children from the United States, Argentina, and Vietnam. The children were longitudinally tested over five periods using the ANT. The authors divided the sample according to the culture of membership (based on the flexibility of social structure), and the results showed that, when other factors (e.g., SES, vocabulary, and age) were controlled for, culture significantly impacted the development of alerting and executive control attentional networks. In contrast, language status was only significant for the executive control attentional network.

### 1.2. Attentional Network and Socioeconomic Status

SES, which is often measured as a combination of education, income, and occupation, has been found to significantly impact attentional function and, more specifically, attentional regulation. For instance, children from low-SES families have been shown to frequently exhibit poor self-regulation [32]—an index of executive control maturity. Studies have also shown that children in middle-SES families tend to outperform their peers in executive control [32,33]. Furthermore, it appears that family SES modifies the mechanism of attention disengagement, which is essential for the development of attention regulation [34]. Several findings have linked low SES to poorer attention regulation to emotional stimuli [34]. In particular, children in low-SES families tend to show diminished activation of the brain structures involved in the voluntary control of attention as adults, when asked to regulate their emotions in response to negative valence stimuli. In addition, infants raised in deprived home environments have been found to show increased negative affectivity, suggesting a link between these variables.

Infants with different temperamental profiles may also be more or less influenced by the environment. One study examined whether stress levels in parents and adolescents were associated with SES and executive functioning in adolescents living in urban areas, finding that parental stress was directly related to adolescent stress and adolescent stress was directly related to the behavioral components of EF (i.e., emotion control, set-shifting, and inhibition) [35]. Finally, Mezzacappa [36] found that children with higher SES were faster and more efficient in the alerting and conflict trials of the children’s ANT than children with lower SES—a finding with direct implications for the present study. In addition, low SES—and particularly a poor-quality home environment—has been shown to indirectly affect children’s inhibitory control and sustained attention [36].

The lack of a systematic measure of SES in previous studies may have led to an overestimation of the bilingual advantage. For example, Ladas, Carrol, and Vivas [37] found that the bilingual advantage was considerably reduced when the monolingual and bilingual participants were matched for SES. Low SES has also been shown to be associated with high levels of stress [38], which can alter cognitive functions, including selective attention. The literature also suggests a possible link between children’s learning environment and the neural correlates of attention. Families with low SES most likely live in chaotic, noisy, and crowded environments [39], and such poor environments have been associated with factors leading to chronic stress, which can affect well-being [40]. Additionally, chaotic environments are unpredictable and inconsistent and are often lacking in routine, which may interfere with healthy development [41].

How do bilingual factors such as the age of learning two languages, language use, and language proficiency affect brain connectivity? The literature suggests that early, lifelong bilingualism alters the structure of gray and white matter in the brain. For example, in studies of adults, greater gray matter (in terms of volume and density) has been found in regions associated with language, such as the bilateral inferior frontal gyrus (IFG), inferior parietal lobule, anterior cingulate cortex, caudate, and putamen [17,41]. In a more recent study involving a different group of highly competent bilingual adults, the same relationship was found in the bilateral IFG. In addition, the authors found that better functional connectivity between cortical and subcortical brain regions, particularly between the left caudate and bilateral superior temporal gyrus, and the anterior cingulate cortex and left putamen, correlated positively with a greater “diversity of language use” (i.e., in an environment where both languages were frequently used and separate use of the languages was not routine) [42]. Overall, there is growing evidence that factors related to bilingualism influence functional connectivity in the brain. However, little is known about how these connectivity changes might relate to executive control performance. In addition, there is growing evidence that bilingualism affects the organization of gray and white matter in the brain. However, a recent study of preschool children (i.e., aged 3–5 years) found no structural changes in the IFG, but more robust functional connectivity in bilingual children than in monolingual children, suggesting that structural changes may manifest only after continuous exposure to two languages [43].

### 1.3. Study Aim

The present study aimed to analyze how SES and bilingualism interact with the attentional network in school-aged children by comparing the performance of bilingual and monolingual third-, fourth-, and fifth-grade children on attentional and cognitive tasks. Specifically, the participants were assessed using an attentional task (i.e., the ANT) that evaluated their attentional alert, orienting, and executive conflict networks, as well as several tests of working memory, short-term memory, and linguistic receptivity. Data were also collected on the families’ SES and the children’s IQ. We chose to analyze the performance of children aged 7–11 years to minimize the risk that the effects of language acquisition would be reflected in the efficiency of the attentional networks.

The above-cited literature suggests that parental SES has a critical impact on children’s cognitive development. However, it is not clear whether this effect might vary as a function of children’s multilingual status. The present study suggests that culture and bilingualism may interact to better explain the previously observed cognitive benefits of bilingualism and the growing discrepancy in the literature over these benefits.

## 2. Materials and Methods

### 2.1. Procedure

The Ethics Committee of the Department of Developmental and Social Psychology at Sapienza University of Rome approved the present study (n.001518).

To recruit participants, we contacted schools with a large multilingual student population. Two schools were selected that were interested in participating. The schools were located in the suburbs of Rome and Naples. A letter was sent to the school principals, the children’s teachers, and the children’s parents or guardians to inform them about the study objectives and procedures. All children were in the third, fourth, or fifth grade, and received parental consent to participate. All assessment sessions took place within the school building in two classrooms that had been reserved for the project on days and at times indicated by the teachers. If a child showed impatience or did not wish to continue, the session was interrupted and continued on another day (4 out of 135 children; 3% of the sample). The teachers acted as intermediaries between the parents and the investigators, who distributed a parent questionnaire to be completed at home. All questionnaires were returned in a sealed envelope, to ensure privacy and anonymity. Data collection began in November 2019 and ended in early March 2020, with the proliferation of COVID-19. Due to the pandemic, some children were unable to complete some tests and were excluded from the study.

### 2.2. Participants

We recruited 135 students, aged 95.05–136.4 months (*M* = 114.6; *SD* = 8.48), who were attending a school on the outskirts of Rome or Naples with a high percentage of international students. The inclusion criteria were that the children had not been diagnosed with a neurodevelopmental disorder and scored higher than 80 on Raven’s colored progressive matrices test. The children’s parents signed an informed consent form and completed a questionnaire that investigated linguistic status (i.e., monolingual, bilingual) and SES. To encourage the children’s participation, at the end of the test sessions, they were given diplomas that certified their participation in the research with the university seal. There were no significant differences between the language groups (i.e., monolingual, bilingual) on non-verbal reasoning, as expressed by the Raven’s colored progressive matrices *z*-scores (*t*(98) = 0.63, *p* < 0.52, *d* = 0.12).

### 2.3. Participants’ Sociocultural and Language Characteristics

SES was assessed using a questionnaire administered to the children’s parents. Participating families lived in a suburb of Rome or Naples, characterized by low or medium-low SES. Most parents had a medium-low level of education, having predominantly earned only a secondary school (father: 39.8%; mother: 41.9%) or middle school degree (father: 34.7%; mother: 28.6%). Few parents had a college degree (father: 7.1%; mother: 9.2%). The parents’ educational level was measured by the number of years they attended school after the age of 5 years. In the seven families with a single mother or father, the single parent’s number of years of schooling was doubled. Monolingual and bilingual children did not differ significantly with respect to their parents’ educational level (*t*(98) = −0.99, *p* < 0.32, *d* = 0.20). We calculated the socioeconomic index of each family by assigning scores to each parent’s years of education and occupation and summing them.

The monolingual children had learned Italian only at home. In the few cases in which one or both parents were non-native Italian speakers, they had lived in Italy for at least 10 years and indicated that their predominant language at home was Italian and their understanding and production of Italian was very good. Their questionnaires indicated that the child had no or very poor understanding of their parents’ first language (L1).

Most bilingual children had been born in Italy (89%) and the rest had settled in Italy within the previous 3 years. These children had learned an L1 other than Italian from birth, and Italian from birth or before the age of 4 years. Only children whose parents reported that their child understood the L1 well or very well were included in the study.

#### 2.3.1. Parent Questionnaire

The questionnaire for parents, “Languages, Speech, and Reading,” investigated the participants’ SES and sociolinguistic conditions. Specifically, the questionnaire examined:-General information about the child and his/her family;-Parents’ occupations and educational attainment;-Parents’ countries of origin and years of residence in Italy;-Languages spoken at home;-Frequency of activities carried out with the child;-Language proficiency in L1 and L2 (reserved for parents with fluency in a language other than Italian).

The parents were asked to indicate for each family member (i.e., mother, father, child, and siblings) which language(s) was/were used in everyday life, choosing among three possible alternatives: only Italian, Italian and other languages (indicating the languages), and only one other language (indicating the language). The questionnaire also investigated parents’ linguistic competence in Italian, ranging from 1 (not at all) to 4 (fluent).

An additional questionnaire for the children was used to obtain information about their use of particular languages at home, at school, and with friends or classmates. Each child was assigned an identification code to ensure anonymity and confidentiality.

#### 2.3.2. Assessment of IQ

Raven’s progressive matrices test (RPM) is a non-verbal test of non-verbal reasoning, abstract thinking, and cognitive ability. Because the test is purely visual, problems of possible language barriers and religious/cultural affiliations are circumvented. Therefore, the RPM is one of the most widely applied measures of general intelligence.

In the RPM, respondents are presented with a 3 × 3 matrix with one part missing. The task is to select the correct diagram from a set of eight options that completes the matrix pattern. The questions and answers are non-verbal, and the matrices differ in the cognitive capacity they require for the correct answer to be identified.

#### 2.3.3. Assessment of Language Abilities

We chose to assess the children’s receptive language, in order to control for their level of language comprehension (which was otherwise only assessed through the parent questionnaire). The Peabody Picture Vocabulary Test [44] is one of the most widely used tests of language ability. It measures receptive processing in subjects aged 2 years and older. The test is administered orally and takes 20–30 min to complete. Respondents do not have to read, and the results are quick to score. To administer the test, the examiner presents the respondent with a series of pictures. One page contains four pictures, each of which is numbered. The examiner says a word that describes one of the pictures and the respondent then points to or says the number of the picture that describes the word. Depending on the age of the respondent, the answers may also be multiple-choice. The total score can be converted into a percentile rank, mental age, or IQ, with standard deviations. No special training is required to administer and score the PPVT-IV. However, the test publisher recommends that anyone seeking to interpret or explain the test results has knowledge of psychological testing and statistics.

#### 2.3.4. Assessment of Working Memory

Non-word repetition. This task comprises part of the Italian VAUMeLF battery [45] (reliability: Cronbach’s alpha = 0.90–0.99). It consists of 40 non-words that are presented via an audio track at 5 s intervals. In the present study, each child’s repetition was scored as 1. The normative battery data converted the final raw score (maximum score: 40) into a *z*-score.

Working memory index (WMI). The WMI was comprised of two subtests of the Italian version of the fourth edition of the Wechsler Intelligence Scale for Children (WISC) [46] (reliability: Cronbach’s alpha = 0.87–0.92): digit span (forward and backward) and letter–number sequencing. In the forward digit span test, the child must repeat numbers in the same order as they were read aloud by the examiner; in the backward digit span test, the child must repeat the numbers in the reverse order as they were read aloud by the examiner. If the child repeats two-digit sequences of the same item incorrectly, the test is terminated. In the letter-– sequence task, the child hears a series of letters and numbers and repeats the stimuli, placing the letters in alphabetical order and the numbers in ascending numerical order (e.g., ‘Please listen to and repeat this series of numbers and letters, first arranging the numbers from smallest to largest and then arranging the letters in alphabetical order: 4-B-1-A→1-4-A-B’). If a child scores 0 on three items, the task is interrupted. In the present study, the children’s raw scores for (a) the forward and backward digit span and (b) letter–number sequencing were first converted into scaled scores, then summed and converted to IQ scores, with reference to Italian normative data for the WISC IV [46].

Immediate narrative memory. In this task, drawn from the Nepsy II battery (reliability: Cronbach’s alpha = 0.69–0.71), the child is asked to listen to a short story and to recall it immediately afterward (free recall score: 0–20). Credit is given for each correct retrieval of a story element, regardless of whether the recall is verbatim, expressed with a similar meaning, or in a different sequence. The test also requires the child to answer open questions about any details that were not spontaneously retrieved (i.e., cued recall) and, eventually, closed questions (i.e., recognition). In the present study, only the free recall score was used. This score, which taps into immediate memory retrieval, was converted to a scaled score (*M* = 10; *SD* = 3), with reference to Italian normative data for the Nepsy II.

#### 2.3.5. Assessment of Attentional Networks

Apparatus. Stimuli were presented on a 12-inch color monitor. Responses were gathered using a standard computer mouse. A PC running E-prime 2.0, Psychology Software Tools (Pittsburgh, PA, USA) controlled the presentation of the stimuli, timing operations, and data collection.

Stimuli. Each participant completed two versions of the ANT, which differed only in the types of stimuli presented. One version presented colored fish as target and flanker stimuli, as described by Rueda et al. [47]. A second version presented photographs of faces, instead of fish. The stimuli and procedure matched those of Federico et al. [48].

### 2.4. ANT Procedure

The experimental session consisted of two tasks: the ANT using fish stimuli (ANT.fish) and the ANT using facial stimuli (ANT.photo). Each task consisted of a practice block of 24 trials and two experimental blocks of 48 trials each. The order of each task was randomized between participants, and participants could take breaks at the end of each practice block and between tasks.

The instructions were the same for all versions of the task. The participants were told that a picture of a face (or fish) would appear on the screen and that they should press the button on the mouse that corresponded to the direction of the gaze of the face (or the direction in which the fish was facing). Each target was preceded by a cue stimulus that either alerted or oriented the participants to the upcoming target. There were four types of cues: (1) no cue (neither an alerting nor an orienting cue was presented), (2) a double cue (a double asterisk cue appeared simultaneously above and below the point of fixation in order to alert the participant), (3) a spatial cue (a single asterisk appeared in the position of the upcoming target, 100% predicting the target position, in order to orient the participant), and (4) a central cue (an asterisk was presented in the position of the fixation cross). The efficiency of the three attentional networks was measured by subtracting the reaction times (RTs) between conditions. The so-called “conflict effect” was calculated by subtracting the mean RTs of the congruent flanking conditions from the mean RTs of the incongruent flanking conditions. The two conditions differed only in the information given by the flankers. When the pictures were congruent, they facilitated discrimination of the target stimulus; in contrast, incongruent flankers distracted the participants.

Visual cues were used to measure orienting and alerting separately. The orienting effect was calculated by subtracting the mean RTs of conditions with spatial cues from the mean RTs of conditions with central cues. Both central and spatial cues alerted the participants to the imminent appearance of the target. However, only spatial cues provided spatial information that allowed the participants to orient their attention to the appropriate spatial location. In the no cue and double cue conditions, attention tended to be divided between the two potential target locations. Neither of these conditions provided spatial information about the location of the target stimulus, but the double cue alerted participants to the imminent appearance of the target. Therefore, the alerting effect (i.e., the benefit of alerting on RT [49,50,51,52,53]) was calculated by subtracting the mean RTs of the double cue conditions from the mean RTs of the no cue conditions.

### 2.5. Statistical Analysis

A multivariate analysis of variance (MANOVA) using linguistic status as the between-group variable, SES as a covariate, and all other variables as dependent variables was conducted to determine any differences between the monolingual and bilingual children and the effect of SES. Each attention network and test score was considered a dependent variable.

## 3. Results

### Linguistic Group and Socioeconomic Status

The results of the MANOVA showed a significant difference between the linguistic groups (*F*(15,120) = 4.92, *p* < 0.0001, η^2^= 0.38) and SES categories (*F*(15,120) = 4.92, *p* < 0.0001, η^2^= 0.38). Table 1 and Table 2 summarize the means and standard deviations of the variables for each linguistic group and SES category, respectively. Table 3 shows the results of the within-group analysis.

Considering linguistic status, the monolingual children performed better in receptive language (i.e., the Peabody Picture Vocabulary Test), even though both groups demonstrated average performance. However, no differences emerged in this variable with respect to SES. Additionally, no effects of linguistic status or SES were observed with regard to working memory (i.e., digit span and letter–number sequencing), short-term memory (i.e., non-word repetition), or narrative memory. Considering attentional networks, in the task involving non-social stimuli, the alerting network was influenced by both linguistic status and SES, with the latter evidencing a stronger effect. The orienting network showed no effect of either bilingualism or SES, while subjects with low SES scored higher in the conflict network. Different effects were found for attentional networks in the task involving social stimuli: a significant effect of SES emerged in the alerting network, but not the orienting network and there was an effect of both linguistic status and SES on the conflict network (Table 3). Of note, in the task involving social stimuli, although the bilinguals showed higher alertness than the monolinguals, the latter showed modulation of this network with respect to SES, with children with the lowest SES status scoring higher (Figure 1).

Interestingly, in the task involving non-social stimuli, bilingual children with low SES scored higher (Figure 2).

With respect to the conflict network, when non-social stimuli were used, both bilingual and monolingual children showed a similar gap in executive system efficiency between those with low and high SES (Figure 3). On the other hand, when social stimuli were used, monolingual children with low SES showed the highest efficiency in executive attention and the lowest conflict score (Figure 4).

## 4. Discussion

There is much controversy regarding whether bilingual individuals generally outperform monolingual individuals in terms of cognitive control [11,22]. According to a recent version of this theory, any advantage may be related to executive attention rather than inhibitory processes, as previously believed [53]. Adding a new element to this discussion, the present study investigated the role of SES in attentional networks in a sample of bilingual and monolingual children.

We chose to analyze the children’s receptive language, working memory, short-term memory, and narrative memory in order to control for the contribution of these cognitive skills to the functioning of attentional networks. In our sample, these systems did not function differently in the monolingual and bilingual children, and they were not significantly influenced by SES. Examining the relationships between the variables, no associations emerged between working memory and SES, linguistic status, or the attentional network. This finding aligns with recent meta-analyses showing no correlation or only a weak association between bilingualism and working memory [26,54,55].

Orsolini et al. produced similar results in their recent study [29] investigating reading comprehension in monolingual and bilingual children. Monolingual children’s comprehension of an oral text was strongly associated with working memory, while bilingual children showed no indirect working memory effect on reading comprehension [29]. As executive attention has historically been closely linked to working memory, this result raises questions about the assumption that multilingualism improves executive attention and that learning a second language expands working memory capacity [56,57].

The most interesting results concern the functioning of attentional networks and, in particular, the alerting and conflict networks. In previous studies, Federico et al. [48,50,51] showed that exposure to real faces positively affected the efficiency of executive control (i.e., the conflict network). Consistent with other studies e.g., [58], we found a bilingual advantage for the attentional network in school-aged children. However, our results also showed that social stimuli seemed to amplify this effect. In fact, in our sample of bilingual children, social stimuli were found to reduce the alerting network, eliminating any differences between high and low SES and strongly reducing conflict, particularly in the bilingual sample with low SES.

Attentional control may be related to social anxiety [59], and this may explain the correlation with SES. In children with high anxiety traits, attentional control is often less efficient than in children with low anxiety traits [60]. In our study, we did not investigate anxiety, but we instead focused on cognitive aspects. However, we are currently collecting data for a new study that will further consider the impact of emotional, social, and family well-being variables in the development of executive function.

Regarding the data presented in this article, the effect of social stimuli on the conflict network may reflect the involvement of sociocognitive aspects that are capable of influencing cognitive functioning.

## 5. Conclusions

The present results suggest that both bilingualism and SES affect attentive networks. In addition, different stimuli may impact attentive performance differently, depending on their social connotations.

Future research in a similar vein could contribute to scientific research on social attention in children. In particular, further studies could focus on training protocols to develop EF abilities in children and specific attention profiles. The current findings contribute to the ongoing dialogue about how bilingualism impacts attentive networks, focusing on the mediating role of SES in attentional skills. Although more investigations are needed, the present results suggest that differences in social contexts, in particular, may affect attentional functioning. Increasing our knowledge of how socioeconomic factors influence different aspects of attention and the ability to utilize higher-order cognitive processes may help us to develop interventions and identify strategies to help children cope with the daily demands of life.

## Figures and Tables

**Figure 1 brainsci-13-00948-f001:**
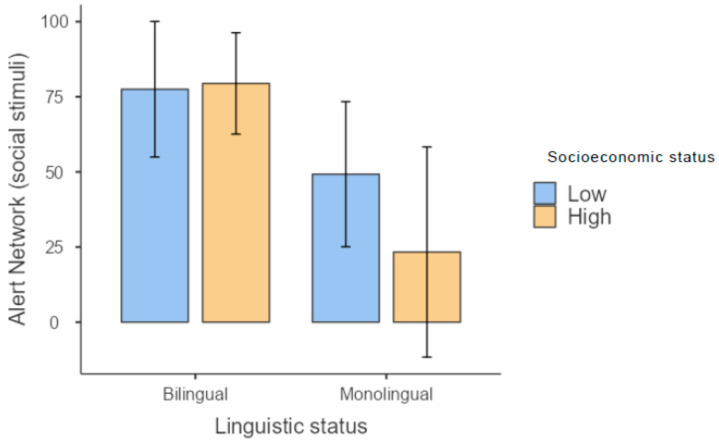
Results for the alerting network in the test involving social stimuli.

**Figure 2 brainsci-13-00948-f002:**
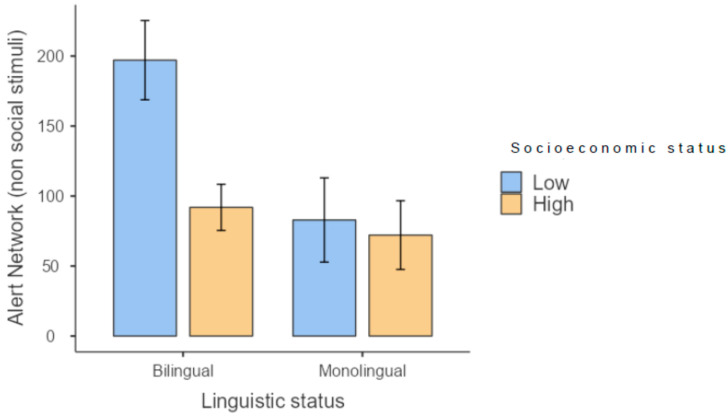
Results for the alerting network in the test involving non-social stimuli.

**Figure 3 brainsci-13-00948-f003:**
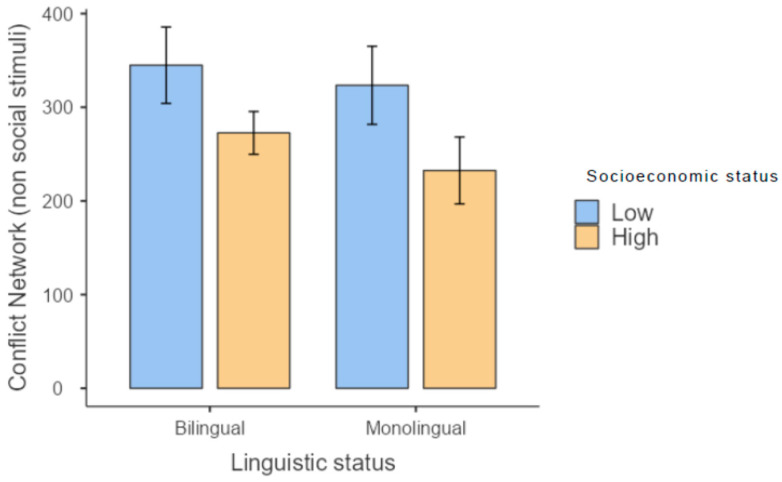
Results for the conflict network in the test involving non-social stimuli.

**Figure 4 brainsci-13-00948-f004:**
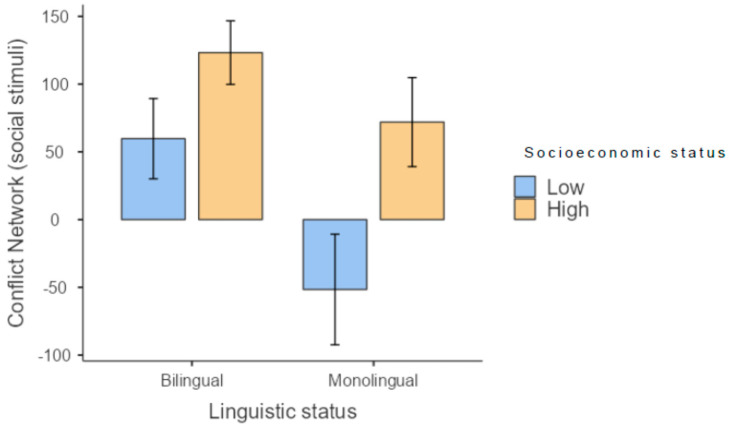
Results for the conflict network in the test involving social stimuli.

**Table 1 brainsci-13-00948-t001:** Means and standard deviations for the tests administered to the bilingual and monolingual subjects.

	Bilingual		Monolingual	
	M	SD	M	SD
Raven’s progressive matrices	−0.13	0.86	−0.16	0.88
Peabody Picture Vocabulary Test	94.09	14.56	96.2	14.88
Digit span	8.57	2.63	8.98	2.81
Non-word repetition	−1.12	1.68	−1.08	1.58
Letter–number sequencing	9.07	3.30	9.3	4.13
Immediate narrative memory	9.65	3.34	10.27	2.89
Alerting photo (rt)	78.81	115.74	32.38	191.92
Orienting photo (rt)	30.54	136.56	45.56	106.24
Conflict photo (rt)	102.62	161.17	28.80	210.97
Alerting fish (rt)	126.03	132.86	75.90	150.73
Orienting fish (rt)	14.37	111.63	12.52	114.21
Conflict fish (rt)	296.09	176.59	264.28	220.50

**Table 2 brainsci-13-00948-t002:** Means and standard deviations for the tests according to high and low SES.

	Low SES		High SES	
	M	SD	M	SD
Raven’s progressive matrices	−0.31	0.89	−0.06	0.85
Peabody Picture Vocabulary Test	94.09	14.56	96.2	14.88
Digit span	8.57	2.63	8.98	2.81
Non-word repetition	−1.12	1.69	−1.09	1.61
Letter–number sequencing	9.07	3.30	9.3	4.13
Immediate narrative memory	9.65	3.34	10.27	2.89
Alerting photo	63.98	111.51	54.16	175.51
Orienting photo	58.05	144.81	27.040	110.34
Conflict photo	6.49	176.11	100.11	187.90
Alerting fish	142.48	149.81	83.01	136.00
Orienting fish	20.83	110.49	9.82	113.80
Conflict fish	334.65	195.81	254.58	194.48

**Table 3 brainsci-13-00948-t003:** Results of the within-group analysis.

	Linguistic Status			SES		
	F	*p*	η^2^	F	*p*	η^2^
Raven’s progressive matrices	0.04	0.843	0	0.367	0.546	0.003
Peabody Picture Vocabulary Test	51.166	0	0.276	0.018	0.895	0
Non-word repetition	0.043	0.837	0	0.762	0.384	0.006
Digit span	0.043	0.837	0	0.079	0.779	0.001
Letter–number sequencing	0.783	0.378	0.006	0.231	0.632	0.002
Immediate narrative memory	1.744	0.189	0.013	1.235	0.268	0.009
Alerting photo	3.386	0.068	0.025	4.086	0.045	0.03
Orienting photo	0.436	0.51	0.003	1.633	0.204	0.012
Conflict photo	5.194	0.024	0.037	13.756	0	0.093
Alerting fish	5.183	0.024	0.037	11.34	0.001	0.078
Orienting fish	0.018	0.893	0	1.013	0.316	0.008
Conflict fish	1.052	0.307	0.008	4.03	0.047	0.029

## Data Availability

I prefer not to share the data now as they are part of a larger dataset still in use.

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
