# Peer review of "Study of Alerting, Orienting, and Executive Control Attentional Networks in Bilingual and Monolingual Primary School Children: The Role of Socioeconomic Status"

_brainsci, 2023, doi:10.3390/brainsci13060948_

Round 1
Reviewer 1 Report
Comments and Suggestions for Authors
The study compares the performance of children grade 3-5, mono- and bilingual, regarding attentional and cognitive tasks. The topic is of high relevance to the readership. However, I have huge and major concerns about several formal and methodological issues as well as about the writing style.
Please see the manuscript for detailed comments. I summarize the main points here:
· Please be clear and consistent with your terminology (e.g., in the manuscript comments)
· The manuscript needs a native-speaker check as it is sometimes hard to understand what is meant. Please avoid too long and complex sentences. The text is not easy to read.
· Please read your manuscript carefully before resubmitting: There are several sentences that are repeated verbatim within the manuscript (see comments in pdf).
· I cannot find an order in the reference list. Please stick to one style (e.g. APA). It is not possible to find the referenced literature. Additionally, the reference list is not complete. Please check on this.
· Introduction:
o The literature review is kind of unordered. Please restructure it.
o Please add more details to all of the reviewed studies in section 1.1. and 1.2., since the reported results are not clear (about what kind of bilinguals are you reporting particular results: Children, adults, exact age, etc.? E.g. p. 3, l.106 “multilingual participants”, etc.
o Please start the section 1.2. by defining socio-economic status. Please insert a statement on what you mean by “attentional network”. Without definitions, it is not clear what exactly is reported here. Please also add information on how SES was operationalized, since this varies widely from study to study. Please be clearer in what exactly was measured in the studies you report.
o The abbreviation ANT is not introduced adequately.
· Section 2:
o This section needs to be filled with much more details on methodological decisions. Please see comments in the manuscript.
o Please be consistent with the use of terminology: non-words or pseudowords?
· Section 3: The results section is not very easy to follow. The table captions cannot be understood. The statistical decisions are not made clear. Why have the authors not chosen to do a regression analysis?
· Section 4: There is much redundancy in the discussion section. Rather than repeating facts, the authors should really discuss their own results and critically reflect on methodological choices, etc. This section has to be rewritten.
Perhaps the authors would like to consider the following literature:
Ehl, B., Bruns, G., & Grosche, M. (2020). Differentiated bilingual vocabulary assessment reveals similarities and differences compared to monolinguals: Conceptual versus single-language scoring and the relation with home language and literacy activities. International Journal of Bilingualism, 24(4), 715-728.
Wimmer, E., & Scherger, A. L. (2022). Working Memory Skills in DLD: Does Bilingualism Make a Difference?. Languages, 7(4), 28,7.

Comments on the Quality of English LanguageThe manuscripts has to be checked by a native-speaker.
Author Response
We apologize for submitting such a confusing and error filled manuscript, we made a mistake while uploading which we only noticed when it was too late. We thank you for having believed in the goodness of our data and for all the work that required such a precise and punctual review. Your comments have been very useful and have helped us to better focus the theme of the work and to rewrite the most problematic sections. Let's hope it's clearer now

Reviewer 2 Report
Comments and Suggestions for Authors
Dear authors, please find the revision of your article below.
Study of attentional networks of alert, orientation, and executive control in bilingual and monolingual primary school children: the role of socioeconomic status
Revue Brain Sciences
The title is clear and promises an original and ambitious study regarding the variables mobilized.
Abstract.
The summary is complete. The controversy is clearly stated. A concluding sentence regarding the possible applications of such a study could be added.
Introduction.
“Attention control is an executive functioning skill critical for operating efficiently in daily life, and gaps in these core processes have severe consequences from childhood to adulthood.” à Could the authors add a scientific reference?
“Since some evidence suggests that bilinguals outperform monolinguals in response times, reflecting attention control (Calabria et al., 2011), and bilingualism is notoriously associated with a cognitive advantage, it seems to be relevant also to investigate the cultural space that provides learning experiences for children to develop cognitive enhancement programs based on these findings, so that they become efficient adults.” à On the contrary, in the summary you say that there is a theoretical controversy about this and that the favorable impact of bilingualism is not unanimous. It would probably be appropriate to close this paragraph with an announcement of the controversy, as this would justify your interest in studying this phenomenon from a new perspective.
1.1. Bilingualism and the attentional network
“Studies demonstrating the benefits of bilingualism in the growth of cognitive processes highlight how speaking two different languages involves the activation of continuous suppression processes for the language that is not being spoken at the time” à Could the authors add a scientific reference?
“According to Bialystok (2011), bilinguals have an advantage in executive functions (EF) because they constantly train to carry on a conversation based on the context and require continued access to the information in the working memory.” à This element is interesting. Could the authors develop the involvement of working memory in a conversational context?
« Some studies found that monolingual-bilingual differences were attenuated when groups were matched on general ability (Antón et al., 2014) or socioeconomic background (Morton & Harper, 2007).” à You say the benefit is mitigated, but does it disappear? This sentence is not clear. Is there a significant effect regardless or not? Do these studies conclude in an advantage of bilingualism, no difference, or a disadvantage?
You say that there are socio-economic disparities between children of bilingual vs. monolingual parents. This is interesting. Could you elaborate? Why or why not?
“These findings show that bilingualism impacts young adults' ability to develop effective attentional processes when that ability is expected to be at its height.” à To conclude that there is a positive impact seems to me to be risky because these are positive correlations, no?
« When other factors (such as socioeconomic status, vocabulary knowledge, and age) are controlled, Tran, Arreondo, and Yoshida (2015) showed that culture significantly impacts the development of alerting and executive control attentional networks. In contrast, language status was only significant for the executive control attentional network. » à I think this study is very important to justify your study. Could you expand on it? Target population? Specific results? Your critical view on the methodology?
1.2. Attentional Network and socioeconomic status.
“For instance, children from low SES families frequently exhibit poor self-regulation (Buckner, Mezzacappa, & Beardslee, 2003), an index of executive control maturity. Measures of executive control also show that children of middle SES outperform their peers (Farah & Noble, 2005; Noble, Norman, & Farah, 2005). Also, it appears that the family’s socioeconomic situation modifies the mechanism of attention disengagement, which is essential for developing attention regulation (Conejero & Rueda, 2018).” à This is very interesting, but how do the authors explain these phenomena?
« More directly relevant to our study, Mezzacappa (2004) found that higher SES children were faster and more efficient in the alerting and conflict trials of the child ANT than lower SES children » à Again, why? How do the researchers explain these results? Is it linked to greater intellectual stimulation in childhood? Is it only genetics?
« Ladas, Carrol, and Vivas (2015) found that the bilingual advantage is considerably reduced when monolingual and bilingual participants are carefully matched on SES. » à Reduced? That's to say ? Not significant effect?
« and families with low SES most likely live in chaotic, noisy, and crowded environments” à Is this an opinion or a scientific fact? Could you add a reference?
1.3. The study’s aim.
What is your general assumption?
2.2. Participants.
I wonder about the choice of this age group. Indeed, the authors were not specified in the theoretical introduction of the age levels of the children targeted in previous studies on the subject. The authors choose children from 8 to 11 years old if I understand correctly. Why this choice? It would be necessary to justify this population about theoretical models of cognitive psychology and developmental psychology.
2.3. Participants’ Sociocultural and Language Characteristics.
Could the authors specify the number of questions asked. The dimensions studied in the questionnaire. The validity of it? If I understand correctly, this is a self-questionnaire, the answers were therefore based on the good faith of the participants. Is this something you questioned? How were the questionnaires distributed? I see you describe the tool in the procedure. I think the tool should be described in the material and stick to how it was administered in the procedure.
Results.
I'm not sure your inferential analysis tables are useful. Also, they are hard to understand. It seems to me that the authors should present the results differently. The problem is that operational assumptions are not made. If this had been the case, the authors could, for each operational hypothesis, carry out a specific analysis to verify it. In my opinion, there are far too many variables to describe the results in this way. The reader does not know what is important or secondary. The authors should rework this section.
Discussion.
The discussion is not really one because the authors do not explain their result. It would be relevant to review the hypotheses, then specify for each of them, if it is verified or not, if it is in line with the literature or not, and if NOT, why? Clearly, what new hypotheses could the authors put forward to justify the innovative result produced by this study. Finally, perspectives should be developed.
Author Response
We apologize for submitting such a confusing and error filled manuscript, we made a mistake while uploading which we only noticed when it was too late. Thank you for your patience and timely review. We followed the directions and practically rewrote the most problematic sections. Let's hope it's clearer now

Reviewer 3 Report
Comments and Suggestions for Authors
The authors examined differences between bi- and monolingual children (language= in several cognitive tasks, considering also socioeconomic status (SES) as a potential factor. They found effects of both language and SES in some variables. Attention was evaluated for social vs. non-social stimuli (stimulus type), which also seems to have had an effect.
I found clear and major flaws in this paper that make it difficult to extract a take-home message. The research question is poorly constructed and even hard to delimit, since there are many dependent variables. Statistical analysis is sometimes floppy, other times absent. The language used is often inaccurate, suggesting poor understanding of statistical objects like types of variables and effects. Basic conventions like reporting results in the past are not respected. Language problems should not be a reason for rejection, but I believe that, altogether, they suggest that the authors might not be able to make the manuscript publishable.
Intro
The introduction lacks focus (e.g., why mention DLD?)
The motivation is hard to understand: why did the authors did all they did - compared bi-and monolinguals for working memory, attention, processing speed, language skills controlling for socioeconomic (or cultural?) status? Why compare social with non-social stimuli?
There are many instances of text that seem to lack sense (e.g., lns 16-17).
Method
There is no section about data analysis, making it hard to understand what guided the authors to present the results they presented (see below)
Results
There are too many levels of analysis whose motivation is not explained. For instance, why correlate dependent variables among them?
Ln 388 - Within-group variables? Aren’t these dependent variables?
Ln 389-…showed a difference …in what?
Table 1. How can we have 94 as z score (Peabody-94 sd above the mean?) How can you have sd of z scores?
Attention-related differences are presented without statistical tests, only with graphs.
Discussion
The discussion does not compensate for all the other problems, and is equally confuse.
Comments on the Quality of English Language
Apart from style-related issues, I found the language poor, mostly for not sounding native-like (not so much because of grammatical errors)
Author Response
We apologize for submitting such a confusing and inaccurate manuscript, unfortunately we made an error uploading the file which we only noticed too late. However following his advice and that of the other reviewers we rewrote the work.
Hopefully this version is clearer
Response to Reviewer 3 comments
Intro
The introduction lacks focus (e.g., why mention DLD?)
Response: We have rewritten the introduction also reducing the reference to DLDs
The motivation is hard to understand: why did the authors did all they did - compared bi-and monolinguals for working memory, attention, processing speed, language skills controlling for socioeconomic (or cultural?) status? Why compare social with non-social stimuli?
Response: we have made the motivation of the study clearer.The comparison between monolingual and bilingual children with high socioeconomic status serves to highlight the different effect of social stimuli in modulating the efficiency of attentional networks with respect to linguistic status and socioeconomic status
There are many instances of text that seem to lack sense (e.g., lns 16-17).
Response: we eliminated the sentences that didn't make sense. We accidentally uploaded the wrong file and we are very sorry for that
Method
There is no section about data analysis, making it hard to understand what guided the authors to present the results they presented (see below)
Response: we have added a section with data analysis
Results
There are too many levels of analysis whose motivation is not explained. For instance, why correlate dependent variables among them?
Response: we have completely restructured the results section
Ln 388 - Within-group variables? Aren’t these dependent variables?
Response: we have completely restructured the results section
Ln 389-…showed a difference …in what?
Response: we have completely restructured the results section
Table 1. How can we have 94 as z score (Peabody-94 sd above the mean?) How can you have sd of z scores?
Response: unfortunately z-score was a typo
Attention-related differences are presented without statistical tests, only with graphs.
Response: statistical tests can be found in the tables
Discussion
The discussion does not compensate for all the other problems, and is equally confuse.
Response: we have completely rewritten the discussion
Reviewer 4 Report
Comments and Suggestions for Authors
The current study evaluated the performance of bilingual and monolingual students in third, fourth, and fifth grades on attentional and cognitive activities. The findings demonstrated that bilingualism and sociocultural status influence attentional networks depending on whether they are stimuli with social connotations or not.
The topic is interesting. The methodology and procedures are adequately described. Figures and tables are useful. The results are well-described.
I would like to make the following minor suggestions:
-I believe that the conclusions should be strengthened with several critical interpretations.
-For the theoretical background, I would also like to suggest Drigas and Mitsea research that links executive functions and attention operations with metacognition.
- Acronyms/Abbreviations should be defined the first time they appear (i.e EF line 32, ADHD line 31).
-According to the journal’s guidelines, references must be numbered in order of appearance in the text. In the text, reference numbers should be placed in square brackets [ ], and placed before the punctuation; for example [1], [1–3] or [1,3].
Author Response
Response to Reviewer 4 comments
I would like to make the following minor suggestions:
-I believe that the conclusions should be strengthened with several critical interpretations.
Response: we have completely rewritten the discussion and the conclusion
-For the theoretical background, I would also like to suggest Drigas and Mitsea research that links executive functions and attention operations with metacognition.
- Acronyms/Abbreviations should be defined the first time they appear (i.e EF line 32, ADHD line 31).
Response: Thanks for your suggestion, and we've improved the presentation of acronyms
-According to the journal’s guidelines, references must be numbered in order of appearance in the text. In the text, reference numbers should be placed in square brackets [ ], and placed before the punctuation; for example [1], [1–3] or [1,3].
Response: thank you, unfortunately we had uploaded a wrong version of the file, now the bibliographic references are numbered correctly
Thank you for your suggestions, we rewrote the manuscript
Round 2
Reviewer 2 Report
Comments and Suggestions for Authors
Thank you
Author Response
Thank you
Reviewer 3 Report
Comments and Suggestions for Authors
The current version of the manuscript is certainly better than the previous one. However, I did not receive a point-by-point response to my comments, so it is difficult to check the authors’ view on these. Nevertheless, I think there are important issues still to b addressed. For example:
-As far as I understood, the literature points to interactive effects of SES and bilingualism (language-related differences reduced when SES is similar) but the table with statistical analysis only considers main effects. Critically, the plots that follow seem to be targeting interactions. Since these are the core results, it would be very important to clarify them.
-within-subjects analysis: the authors refer to it when comparing for language and SES. I think this is not within, but between subjects. Within would be comparing performance across tests.
Also, I think the authors did not have enough time yet to prepare the manuscript:
-we see repeated paragraphs on page 3.
-we see comments from someone (was it a Reviewer?)
I would also like to learn about the authors' views on my previous comments (first round), since I may be getting something wrong.
Comments on the Quality of English LanguageThe language has been improved.
Author Response
Dear Reviewer 3,
many thanks for your useful help.
1) As far as I understood, the literature points to interactive effects of SES and bilingualism (language-related differences reduced when SES is similar) but the table with statistical analysis only considers main effects. Critically, the plots that follow seem to be targeting interactions. Since these are the core results, it would be very important to clarify them.
Response: Our graphs were made using descriptive statistics, in the tables we report the effect of the analysis of variance between bilinguals and monolinguals using ses as a covariate
within-subjects analysis: the authors refer to it when comparing for language and SES. I think this is not within, but between subjects. Within would be comparing performance across tests.
Response: We make a mistake, we changed the text
we see repeated paragraphs on page 3.
Response: I can't see the repetition
-we see comments from someone (was it a Reviewer?)
Respose: thanks for reporting, we have removed them
you can find attached the point x point response to the previous round